# A Meta-Reinforcement Learning Algorithm for Causal Discovery

Andreas W.M. Sauter[1]        Erman Acar[1, 2]        Vincent François-Lavet[1]

[1]Computer Science Dept., Vrije Universiteit Amsterdam, The Netherlands
[2]LIACS, Universiteit Leiden, The Netherlands

## Abstract

Causal discovery is a major task with utmost importance for machine learning since causal structures can enable models to go beyond pure correlation-based inference and significantly boost their performance. However, finding causal structures from data poses a significant challenge both in computational effort and accuracy, let alone its impossibility without interventions in general. In this paper, we develop a meta-reinforcement learning algorithm that performs causal discovery by learning to perform interventions such that it can construct a causal graph. Apart from being useful for possible downstream applications, the estimated causal graph also provides an explanation for the data-generating process. In this article, we show that our algorithm estimates a good graph compared to the SOTA approaches, even in environments whose underlying causal structure is previously unseen. Further, we make an ablation study that shows how learning interventions contribute to the overall performance of our approach. We conclude that interventions indeed help boost the performance, efficiently yielding an accurate estimate of the causal structure of a possibly unseen environment.

## 1 MOTIVATION AND CONTRIBUTION

From daily routines to scientific investigations, many questions can be broken down into causal questions like "Why does this error message constantly appear on my screen?" or "Does more physical activity reduce the risk of cardiovascular diseases?". As solving such a task requires strong *generalisation* and *transfer* (of knowledge) skills, it also stands as one of the challenges with utmost importance in machine learning (ML) research [Schölkopf et al., 2021].

Causal discovery is the task of finding the causal structures

of environments, given some data about their observable variables. These structures enable models to go beyond pure correlation-based inference and significantly boost their performance. This is why there has been a recent push for causality-driven ML. This drive also goes the opposite way, where ML supports causality research to deal with and draw conclusions from large amounts of data [Peters et al., 2017, Schölkopf et al., 2021, Guo et al., 2020]. To infer causal structure from data it has been shown that it is necessary to perform *interventions* [Pearl, 1993, Bareinboim et al., 2020]; that is to experimentally 'force' a variable to take on a certain value. Only these interventions allow us to, in general, distinguish causal structures which yield the same observational distribution over their variables.

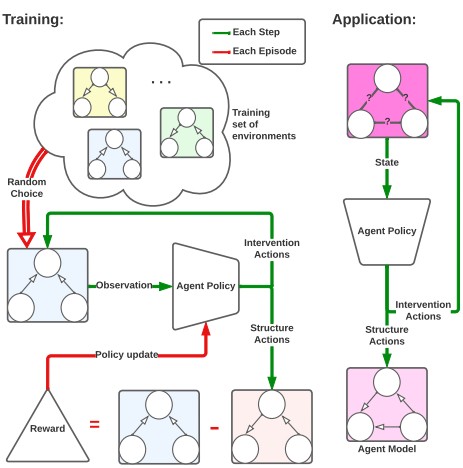

Figure 1: Our policy learns to intervene and estimate causal structures during training. It can then be applied to environments that have a structure unseen during training.

We develop a meta-learning algorithm in a reinforcement learning (RL) setting where the agent learns to intervene to construct a causal graph. An overview is given in Figure 1. Find the code at https://github.com/sa-and/

*Accepted for the Causal Representation Learning workshop at the 38th Conference on Uncertainty in Artificial Intelligence* (UAI CRL 2022).

The research field of causal discovery has led to many techniques. Constraint-based algorithms (e.g. PC and FCI [Spirtes et al., 2000]) infer the causal relationships through independence tests on purely observational data. Score-based algorithms (e.g. GES [Chickering, 2002, Meek, 1997], FGS [Ramsey et al., 2017], GIES [Hauser and Bühlmann, 2012]) incrementally add and delete edges such that the structure of the causal model is improved w.r.t. some scoring function. Another line of research searches over the space of permutations rather than a graph space (e.g. GSP [Solus et al., 2017], IGSP [Wang et al., 2017a]) with extensions even to soft interventions [Yang et al., 2018]. Recent papers also make use of ML and RL methods to discover causal graphs. Such works range from graph-generating neural networks (e.g. CGNNs [Goudet et al., 2018, Ton et al., 2020]) to methods that use encoder-decoder models [Yu et al., 2019] with a search guided by RL [Zhu et al., 2019] to constraint optimization (NOTEARS [Zheng et al., 2020]). We point out another paper by Dasgupta et al. [Dasgupta et al., 2019] that is not directly aimed at causal discovery. Rather, using RL, it solves a prediction task that depends on the ability of the model to learn causal effects from interventions in the environment.

Most causal discovery algorithms come with challenges and shortcomings. Almost all aforementioned approaches can not make use of interventions at all or are computationally inefficient. This is largely due to their inability to use previous information to generalise to unseen environments. In this work, we tackle these challenges by implementing a meta-RL algorithm that learns to learn causal structures using interventions. Our main contributions tackle two main common issues found in causal discovery, briefly formulated as follows:

(I1) computationally efficient use of interventions,

(I2) generalization to environments with unseen and unknown causal structure

In this context, we evaluate our approach by carrying out a series of experiments and show how it compares to existing approaches in varying sizes of environments. We then make an ablation study to understand how much the interventions contribute to our performance. It turns out that our approach compares favourably to SOTA approaches regarding the aforementioned issues.

## 2 PRELIMINARIES AND NOTATION

Causal relationships are formally expressed in terms of a *structural causal model* (SCM). Every SCM induces a graph structure $G$ in which each node and edge represent a random variable and direct causal effect between nodes, respectively. We define an SCM $S$ as a tuple $(\mathcal{X}, \mathcal{U}, \mathcal{F}, \mathcal{P})$

where $\mathcal{X} = \{X_1, \ldots, X_{|\mathcal{X}|}\}$ is the set of *observable* (also called *endogenous*) variables; $\mathcal{U} = \{U_1, \ldots, U_{|\mathcal{U}|}\}$ is the set of *unobservable* (also called *exogenous*) variables; $\mathcal{F} = \{f_1, \ldots, f_{|\mathcal{F}|}\}$ is the set of functions whose elements are defined as *structural equations* in the form of $X_i \leftarrow f_i(Pa_{X_i}^G, U_i)$ in which $Pa_{X_i}^G$ is the set of observable parents of $X_i$ w.r.t. $G$; $\mathcal{P} = \{P_1, \ldots, P_{|\mathcal{U}|}\}$ is a set of pairwise independent distributions where $U_i \sim P_i$. Moreover, we will assume that unobservable variables do not have any parents in $G$ i.e., every $U_i$ is a root node.

We will assume the induced causal graph is always a *directed acyclic graph* (DAG) i.e., acyclic SCM. We will also make use of the notion of a *partially directed acyclic graph* (PDAG) which can be thought of as a DAG where some edges are relaxed to be bi-directional, and cyclicity is only limited to those edges.

A intervention[1] on a variable $X_i$ is defined as replacing the corresponding structural equation $X_i \leftarrow f_i(Pa_{X_i}^G, U_i)$ with $X_i \leftarrow x$ for some value $x$, which we denote as $do(X_i = x)$. Intervening makes the variable independent of its parents, changing the causal mechanism of the data-generation process. The model is causal in the sense that one can derive the distribution of a subset $\mathcal{X}' \subseteq \mathcal{X}$ of variables following an intervention on a set of variables, called *intervention target*, $\mathcal{I} \subseteq \mathcal{X} \setminus \mathcal{X}'$. We call the resulting distribution over $\mathcal{X}$ *post-interventional*. When no intervention is performed ($\mathcal{I} = \emptyset$) we will call it the *observational* distribution.

## 3 WORKING ASSUMPTIONS

Before a detailed description of our approach, we state some assumptions made in this paper. We make these assumptions mainly for the simplification of the algorithm. The main assumptions are the following:

(A1) Each environment is defined by an acyclic SCM.

(A2) Every observable variable can be intervened on.

(A3) For each environment in the training set, the underlying SCM is given.

(A4) We perform interventions on at most one variable at a time.

## 4 REINFORCEMENT LEARNING SETUP

### 4.1 ACTIONS

We implement two types of discrete actions. The first type performs an intervention on the environment and observes the resulting values of the variables. This enables the policy to choose a (post-interventional) distribution and to sample from it. We will refer to this kind of action as *listening action*.

---

[1] In this work we only consider so-called hard interventions.

All, except for one, of the listening actions are *intervention actions* that intervene on exactly one variable (i.e., $|\mathcal{I}| = 1$). Inspired by [Dasgupta et al., 2019], for each observable variable $X \in \mathcal{X}$, we provide an action $do(X = 0)$ and $do(X = 5)$. This results in a total of $2n$ intervention actions for $n$ nodes. There is one additional listening action which we call the *non-action*. When the non-action is taken, the agent observes the current values of the observable variables without intervening (i.e., $\mathcal{I} = \emptyset$). This action accounts for the collection of purely observational data.

The second type of action is responsible for constructing the *epistemic model* of the causal structure of the environment, which is the current best (PDAG) estimate of the agent. We will refer to these actions as *structure-actions*. Each structure action can either *add*, *delete* or *reverse* an edge of the epistemic causal model.

For a graph with $n$ nodes, there are $n(n-1)$ possible edges, and hence there are $3n(n-1)$ structure-actions. Together with the listening-actions we have $2n + 1 + 3n(n-1) = 3n^2 - n + 1$ actions. So the size of the action-space is quadratic in the size of nodes. Whenever a delete or reverse action is applied to an edge that is not present in the current model, the action is ignored. This is effectively equivalent to performing the non-action. The same holds when the add action is applied to an edge that is already in the epistemic model. We do not make any further restrictions, for instance, w.r.t. acyclicity for the structure actions.

## 4.2 STATE SPACE

The state of the environment consists of a concatenation of three parts. The first one encodes the current values of the $n$ observable variables. The second part is a one-hot encoding of which variable is currently being intervened on. The third part of the state encodes the current epistemic model as a vector. Each value of this vector represents an undirected edge in the graph. The edges in the vector are ordered lexicographically. The value 0 encodes that there is no edge between the two nodes. The value 0.5 encodes that there is an edge going from the lexicographically smaller node to the bigger node of the undirected edge. And the value 1 encodes that there is an edge in the opposite direction.

## 4.3 REWARDS AND EPISODES

Our task is to find the causal structure of the environment, i.e., the DAG that corresponds to the graph induced by the SCM. Therefore, we compare the PDAG generated by the agent in an episode to the true causal structure of the environment. The quantification of this comparison serves as the reward for our algorithm.

One obvious choice is to count the edge differences between the two graphs. This ensures that generating a model that has

more edges in common with the true DAG will be preferred over one which has fewer edges in common. It further gives a strong focus on causal discovery as opposed to scores based on causal inference. The *Structural Hamming Distance* (SHD) [Tsamardinos et al., 2006] provides a metric that describes a way of counting the differences between two directed graphs. In our case, it takes two PDAGs and counts how many of the following operations are needed to transform the first PDAG into the second: add or delete an undirected edge, and add, remove, or reverse the orientation of an edge. For implementational purposes we adapt this metric to not count reverse actions as such but by counting them as add and delete actions instead. Further, we count undirected edges[2] as bi-directed edges. This results in a metric that simply counts the distinguishing edges of two directed graphs. We will refer to this metric as *directed SHD* or *dSHD*. Given a predicted directed graph $S_P = (V, E_P)$ and a target, directed graph $S_T = (V, E_T)$, we define the dSHD as $dSHD(E_P, E_T) = |E_P \setminus E_T| + |E_T \setminus E_P|$.

As we need to determine when the estimation of the model terminates, we set a finite horizon $H$ for each episode. The estimation of the epistemic model is complete when $H - 1$ actions were taken. Dynamically determining the end of the estimations is left for future research. This is also the only time the quality of the causal model is evaluated. Note that when a small episode length is chosen, there might not be enough steps available to the agent to collect enough data and make the right changes to the epistemic model. We suggest that the episode length should be at least $n + n(n-1)/2$, which allows for one intervention on each node and one operation per possible edge in the graph. Further, $H$ should not be set too large since additional learning complexity might be introduced. At the beginning of each episode, an SCM is sampled from the training set. The epistemic causal model of the agent is reset to a random PDAG, to further introduce randomness. The evaluation is done by calculating the negative dSHD between the generated PDAG and the true causal graph only at the end of each episode. Every other step receives a reward of 0. The value function for a state $s$ and a policy $\pi$ is then defined as

$$V_\pi(s) = \mathbb{E}_\pi \left[ -\gamma^H \text{dSHD}(edges(s_H), E_{Env}) \mid s_t = s \right]$$

where

- *edges(s)* is the operator that returns the edges of a state,
- $E_{Env}$ are the edges of the current target graph,
- $H \in \mathbb{N}$ is the horizon.
- $\gamma$ is the discount factor $\gamma \in [0, 1]$.

We use the *Actor-Critic with Experience Replay* (ACER) [Wang et al., 2017b] algorithm to solve this RL problem. We choose this algorithm because it is a sample-efficient

---

[2]Graphs generated with other algorithms might contain undirected edges.

off-policy methods and its (potential) easy extension to continuous action spaces. We use a discount factor $\gamma = 0.99$ and a buffer size of 500000. All other parameters are according to the standard values of the Python library we used (Stable-Baselines 2.10.1 [Hill et al., 2018]).

## 4.4 POLICY NETWORK

Both, the actor-network and the critic network are fully-connected MLPs. Both networks are preceded by a shared network that has $n$ fully-connected feed-forward layers followed by a single LSTM layer. The exact amounts of layers and their sizes are specified for each experiment.

We want to emphasise the recurrent LSTM layer. It enables the policy to memorize past information of its preceding layers and, therefore, use information from previous observations. More specifically, it should enable the policy to remember samples from the (post-interventional) distributions induced by the data-generating SCM earlier in that episode. We argue that this should help to better identify causal relations since the results of sequential interventions can be used to estimate the distribution.

## 5 LEARNING TO INTERVENE

First, we develop a toy example to test whether our approach can learn to perform the right interventions to identify causal models under optimal conditions. To this end, we construct a simple experiment in which two observationally equivalent, yet interventionally different environments have to be distinguished. This can only be achieved with the help of interventions [Bareinboim et al., 2020]. Thus, if our policy learns to distinguish those environments, it has to learn to perform interventions. The two environments are governed by the fully observable, 3-variable SCMs with structures $G_1 : X_1 \leftarrow X_0 \rightarrow X_2$ and $G_2 : X_0 \rightarrow X_1 \rightarrow X_2$. In both environments, the root node $X_0$ follows a normal distribution with $X_0 \sim N(\mu = 0, \sigma = 0.1)$. The nodes $X_1$ and $X_2$ take the values of their parents in the corresponding graph. The resulting observational distributions $P_{G_1}(X_0, X_1, X_2)$ and $P_{G_2}(X_0, X_1, X_2)$ are the same and so are the post-interventional distributions after interventions on $X_0$ or $X_2$. For an intervention on $X_1$, $P_{G_1}(X_0, X_2 \mid do(X_1 = x)) \neq P_{G_2}(X_0, X_2 \mid do(X_1 = x))$. Hence the two SCMs can only be distinguished by intervening on $X_1$.

The parameters for this experiment can be found in Appendix A.1. The algorithm is trained in both environments. This allows us to investigate whether, given enough training time and data, our approach *can* learn to distinguish the environments. During training, we observe that the mean dSHD of the produced graphs is 0.0 with a standard derivation of 0.0. This is a perfect reproduction of the two environments in all cases. This indicates that our policy has learned to

use the right interventions to find the true causal structure. After training, we apply the converged policy 10 times to each of the environments and qualitatively analyze the behaviour. What the resulting 20 episodes have in common is that, towards the beginning of each episode, they tend to delete edges that do not overlap in the two environments. Then an intervention on $X_1$ is performed. Depending on the outcome of the intervention, either $G_1$ or $G_2$ is ultimately generated. This can also be seen in the example in Appendix A.2.

This shows that our learned policy learns to use the intervention on $X_1$ to distinguish between the two environments. Thus, our approach is capable of using interventions in a step-wise manner to perform causal discovery. Furthermore, these results suggest that the model has learned to only perform interventions that are relevant as opposed to random interventions. This constitutes a step toward solving the common issue of the efficient use of interventions (I1).

## 6 GENERALISATION OF CAUSAL KNOWLEDGE

We now investigate how well a policy can transfer its knowledge about causal discovery from a set of training environments to unseen test environments. We benchmark our learned policy against established methods.

Following the widely adopted practice, we test our approach on environments that are fully observable and have an additive linear causal model with Gaussian noise. It is known that these kinds of environments suffer from varsortability where good results can be achieved by ordering the variables by the variance of their observational distribution [Reisach et al., 2021, Kaiser and Sipos, 2021]. Since our agent successfully leverages interventions (see Section 5 and 7) for estimating causal structures, we argue that this effect can only partly account for the performance. A more detailed investigation is left for future research.

Given a structure $G$ with a set of observable variables $\mathcal{X}$, we model our environments as $\forall X \in \mathcal{X}$ :

$$X \leftarrow \left( \sum_{Y \in Pa_X^G} WY \right) + \varepsilon \qquad (1)$$

where $\varepsilon \sim N(\mu = 0, \sigma = 0.1)$ is some random noise and $W \sim \texttt{Uniform}(\{\pm 0.2, \pm 0.4, \pm 0.6, \pm 0.8, \pm 1\})$ represents a random weight for each causal effect of a parent to a child.

We create 24 SCMs with 3 observable variables and 542 SCMs with 4 observable variables. The SCMs in these sets all induce distinct causal structures. The exact training setup can be found in Appendix B. We will refer to the model which performed best on the test set as *best model*.

We created a random baseline that returns a random DAG. To compare the learned policies to established algorithms, we ran our best model and the other approaches (NOTEARS [Zheng et al., 2018, 2020], GES [Chickering, 2002, Kalainathan and Goudet, 2019], random) 20 times on each of the environments in the test sets and computed the directed SHD for each of the generated graphs. For NOTEARS and GES, 1000 random samples from the observational distribution of each environment are taken as input. Figures 2 and 3 show the boxplots of the average directed SHD of the 20 runs on each of the environments in the test set.

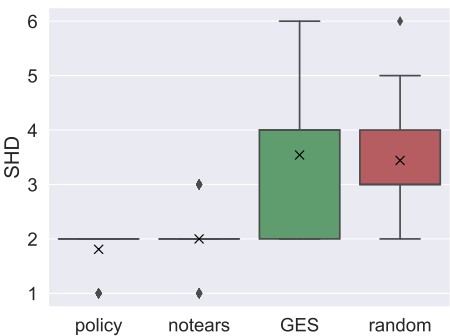

Figure 2: Boxplots of mean dSHD over 20 runs on each environment of our policy, NOTEARS, GES, and our random baseline on the test set for 3-variable environments. The 'x' indicates the mean.

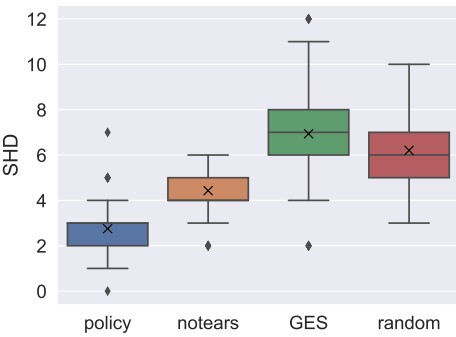

Figure 3: Boxplots of mean dSHD over 20 runs on each environment of our policy, NOTEARS, GES, and our random baseline on the test set for 4-variable environments. The 'x' indicates the mean and the bar in the middle of the box is the median.

Figure 2 shows that our approach outperforms the random baseline, suggesting that our policy learns to estimate the environment's causal structure beyond randomly orienting edges. Furthermore, we can see that our policy outperforms the GES algorithm which is based on purely observa-

tional data. As shown by Zheng et al. [Zheng et al., 2018], NOTEARS outperforms FGS which is based on GES. This can also be seen in Figures 2 and 3. Such relative higher performance is also the reason why we focus on comparing our approach to NOTEARS. The boxplots suggest that the dSHD of our approach compares favorably to the one of NOTEARS. To investigate this difference in more detail, we performed a Wilcoxon signed-rank test between the directed SHDs resulting from our policy and the ones resulting from running NOTEARS. To ensure that the assumption of independent samples holds, we only compare the result of the first evaluation of each environment. In the 3-variable environments, the median of the directed SHDs of our approach *is not* significantly lower than the one from NOTEARS with $p \approx 0.159$ (Note that this is based on only 5 samples for each algorithm). In the 4-variable environments, however, the median of our approach is significantly better than NOTEARS (with $p \approx 4.8 \cdot 10^{-7}$).

We conclude that applying our learned policy to these environments can be preferable to applying NOTEARS or GES. This also suggests that our model can generalise to previously unseen environments. One possible explanation is that the quality gain w.r.t. NOTEARS and GES is attributed to the fact that they are both based on purely observational data, whereas our policy leverages interventional data.[3] We investigate how big the influence of interventions is in our approach in the next section.

## 7 CONTRIBUTION OF INTERVENTIONS

To empirically investigate the effect of interventions on the performance of our algorithm, we perform an ablation study. We train a variant of our policy which is based on purely observational data (we disallow the use of interventions) and compare it to the model which uses interventions. We then compare our results again to NOTEARS which is also using purely observational data.

We measure the average directed SHD of the training set as well as the test set of the current model every 500000 steps. We take the model which performed best on the test set and run it 20 times on each environment of the test set and measure the dSHD for every generated graph. We do the same for the best models of the previous section and the NOTEARS algorithm. We then proceed by doing a Wilcoxon signed-rank test to evaluate whether there is a significant difference between the model that uses interventions and the one that does not. We also test whether there is a difference between the NOTEARS algorithm and our approach when no interventions are allowed. Note that we perform the test only on the first run for each environment to ensure the independence of the samples at the cost of

---

[3]A comparison to *interventional* SOTA approaches is left for future research.

yielding small sample sizes.

Figure 4 shows the learning progress of our model on 4-variable environments. We can see in the first part, that the average dSHD on the test and training of both models compares similarly. What stands out is that after approximately 10 million training steps, the performance of the model without interventions turns out to be approximately constant. At about the same amount of training steps, the model which uses interventions starts to rapidly perform better until reaching almost 30 million training steps in total. For the model with interventions, a learning process emerges with two fast learning regions. We argue that this behavior emerges from a two-phase learning process. First, the model learns to produce a graph, based on purely observational data. And then, they learn to use interventions (if enabled) to further increase performance.[4]

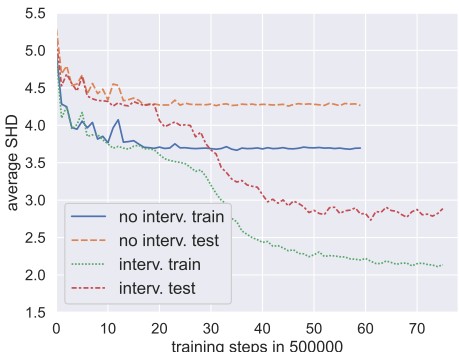

Figure 4: Learning progress of an agent based on purely observational data (blue, orange) and one that can perform interventions (green, red) on 4-variable environments.

We also compare the three models statistically. Here, the median dSHD of the graphs produced by the best model with interventions is significantly lower than the one of the best models without interventions with a $p$-value of $\approx 4.8 \cdot 10^{-8}$. Furthermore, the median dSHD of the produced graphs of the model without interventions is significantly lower than the ones produced by NOTEARS with a $p$-value of approximately $0.004$. This means that even in the case of purely observational data, our approach can outperform NOTEARS in 4-variable environments. Similar results can be found for the 3-variable environments as is shown in Appendix C.

This leads to the conclusion that introducing interventions results in the hypothesized edge over the purely observational version of our model. Ultimately, in the 4-variable environments, our model outperforms NOTEARS no matter whether it uses interventional data or purely observational

one. Nonetheless, the model using interventional data outperforms the one using purely observational data. Overall, these empirical results support the idea that interventions help to identify causal structures from data.

# 8 DISCUSSION AND CONCLUSION

We proposed a new meta-learning algorithm in RL setting for the task of causal discovery. Our exploratory experiments showed that our policy can learn to perform interventions that are strongly informative for the current environments (I1). We quantified the quality of the estimated models on previously unseen environments with 3 and 4 variables. We showed that our policy generalises well also w.r.t. established algorithms based on observational data (I2). In an ablation study, we elaborate on the role that interventions play in the good performance of our approach.

The related works from the active learning community such as Murphy [2001], Scherrer et al. [2021], Tigas et al. [2022] and Amirinezhad et al. [2022], we have only come to know after the reviewers have pointed them out. Of those works, the closest to ours is Amirinezhad et al. [2022] which employs a reinforcement learning approach as well, while focusing rather on minimising the number of interventions. In doing so, as an architectural difference, they employ both a graph neural network to learn the embedding vector of each node and an additional network that scores the output of the embedding network to choose the node to be intervened. Then they apply intervention, and the Meek rules to orient and reiterate the process. For a fair comparison, benchmarking against these works and thorough comparative analysis are needed and are parts of our future agenda.

Furthermore, we acknowledge that our approach needs modifications to scale to realistic environments with more variables. The resulting explosion of the action- and state-space needs to prompt considerations about better encodings. A further problem in a real-world setting is the availability of a large amount of data-generating models for training. These kinds of models are often unavailable. Another issue arises from the assumption that every observable variable can be an intervention target (A2). In real-world settings, causal variables might not be accessible by interventions (e.g., outside temperature) or the abstraction level of an intervenable variable is unclear. For instance, it might be that the input pixels are not the variables we can intervene on but the color of a light bulb which is represented by those pixels is. Future research can be done on an extension to intervention targets of arbitrary size (A4). Also, the possibility of transfer learning between different classes of causal models, such as those not governed by a linear additive model, can prompt interesting investigations. Lastly, we point out that, to fully evaluate our approach, an extensive benchmark against SOTA interventional algorithms has to be conducted in future research.

---

[4]Note that we trained the interventional model two additional times where it exhibited such two-phase behavior as well. For visualization purposes, we omitted these runs in the plot.

## ACKNOWLEDGEMENTS

Thanks to all the reviewers for their constructive and insightful comments which helped this manuscript with a substantial improvement.

This research was partially funded by the Hybrid Intelligence Center, a 10-year programme funded by the Dutch Ministry of Education, Culture and Science through the Netherlands Organisation for Scientific Research, https://hybrid-intelligence-centre.nl, grant number 024.004.022.

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

# A  EXPERIMENTAL DETAILS FOR PROOF OF CONCEPT

## A.1  MODEL PARAMETERS

The policy network for the experiment in Section 5 has a fully connected layer of size 30, followed by an LSTM layer of size 30. The actor-network has one fully connected layer of size 30, the critic-network one fully connected layer of size 10. The length of each episode was set to 10 and the model trained for 5 million training steps. For all other parameters, the default values were used.

## A.2  ILLUSTRATION

Here we illustrate two hand-picked episodes, one for each environment, to render a better picture of how the graphs are produced. These two episodes can be seen in Figure 5. Each episode shows an application of the fully converged policy on the environments as described in Section 5.

# B  TRAINING SETUP FOR GENERALIZATION EXPERIMENT

The following configuration for the policy network worked best after preliminary experiments for the 3-variable (4-variable) environments: One (two) fully connected layer(s) of size 30 (40) followed by an LSTM layer of size 30 (160). Its outputs are fed into a fully connected layer of size 30 (60) for the actor-network and one of size 10 (20) for the critic-network. For this experiment, we set the episode length to 20. The network was evaluated every 500000 steps in both cases. All runs were stopped after 14 evaluations of the policy on the test setwere worse than the best policy found so far to avoid overfitting. In the 3-variable set, we use the first 18 environments for training and the last 5 for testing. In the 4-variable set, we use the first 500 environments for training and the last 42 for testing. The best model for the 3-variable (4-variable) environments is obtained after 14.5 (31) million training steps.

# C  INFLUENCE OF INTERVENTIONS ON ENVIRONMENTS WITH 3 VARIABLES

Considering the influence of interventions on 3-variable environments, the results, at first, look slightly different as see in Figures 6 and 7. The figures suggest that the interventional model does not use interventions when applied to the test set. Fortunately, this visual inspection is misleading as the Wilcoxon rank-sum test on the runs of the models on the test set shows. Here, the directed SHD of the graphs produced by the interventional model has a lower median than its purely observational counterpart. With a $p$-value

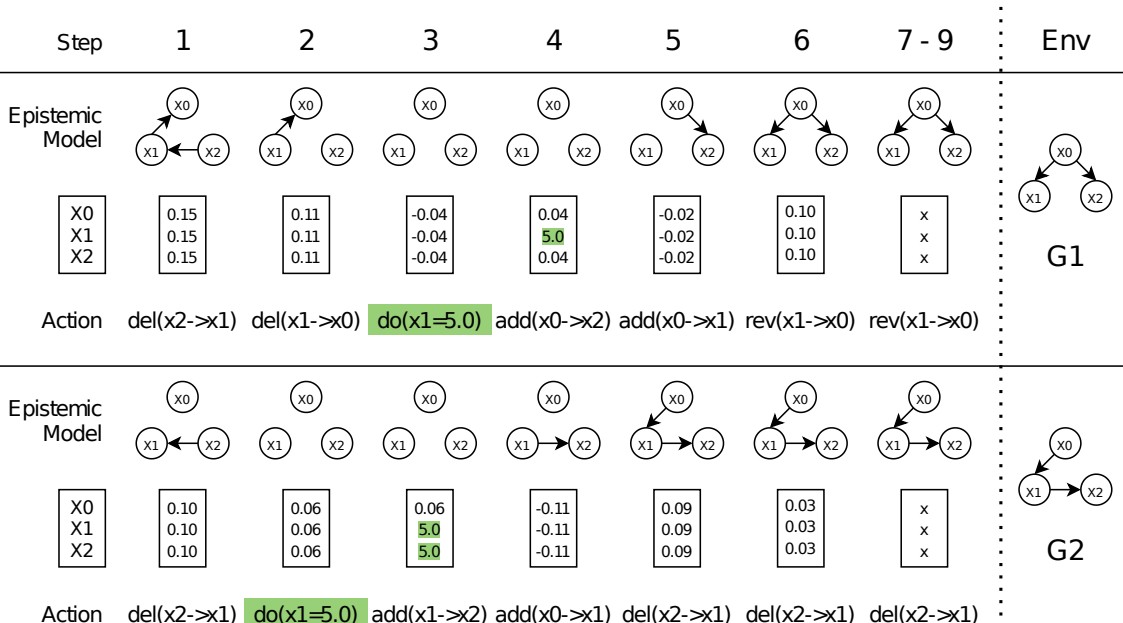

Figure 5: Illustration of two sample episodes after training with the respective causal environments $G_1$ and $G_2$. Interventions and their effects are highlighted.

$\approx 0.0156$, this is slightly significant even at a Bonferroni-corrected significance level of 0.025 (mind, again, the small sample size of 5). At the same time, there is no significant difference in the medians of the no_intervention condition and the NOTEARS condition when performing a two-sided Wilcoxon rank-sum test ($p \approx 0.063$).

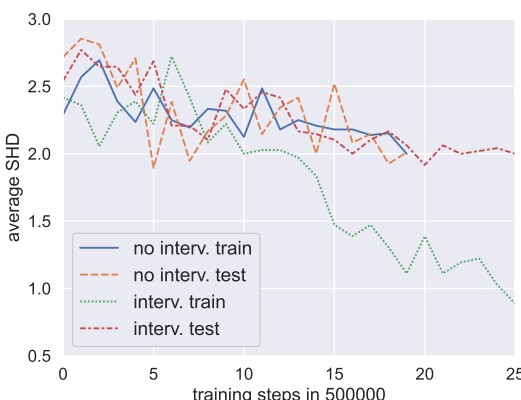

Figure 6: Learning progress of an agent based on purely observational data (blue, orange) and one that can perform interventions (green, red) on 3-variable environments.

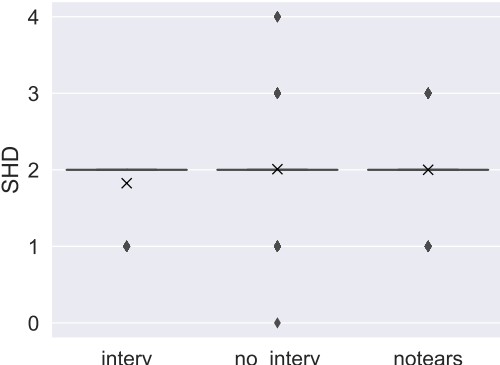

Figure 7: dSHD of the three models for 3-variable environments on the test set.