# OpenReview forum: "A Meta-Reinforcement Learning Algorithm for Causal Discovery"
_auai.org/UAI/2022/Workshop/CRL — CRL@UAI 2022 Poster_

### Official Review · Reviewer_7fRz · 2022-06-25
**Interesting work that bridges causal discovery and meta RL**

**Rating:** 6
**Confidence:** 4

**Review:**

Overall, the proposed idea that formulates causal discovery with active learning in a meta RL setting is interesting and novel (to the best of my knowledge). Furthermore, this paper is a timely contribution that tries to bridge the gap between causal discovery and modern ML techniques (particularly meta RL), which complements other recent causal discovery methods based on differentiable optimization, deep learning, reinforcement learning, etc. To my knowledge, the proposed method appears to be sound. I would encourage the authors to take the following comments into account when preparing for future revision.

Comments about the method:
- How does the agent ensure that a directed cycle does not appear in the final solution?
- In Section 4.1, two of the actions provided are $do(X=0)$ and $do(X=5)$. The particular choices seem adhoc to me--could the authors provide more explanation about that? Could the intervened values be part of the action, say $do(X=k)$ where $k$ is some value to be learned by the agent as well?
- Apart from SHD, it might be interesting to consider using SID [1] as reward as a future work, which is more appropriate for the causal inference setting.

Comments about clarity and writing:
- The paper will benefit from giving a proper definition of "environment". If I understand correctly, the authors are using "environment" to refer to that of the RL or meta RL setting, which may be different from an "environment" (or "domain") in the causal discovery setting, e.g. [2].
- I find it difficult to follow from Section 3 directly to Section 4.1. In between them, the authors should include a clear description of the problem setup and a brief overview of the proposed framework (it is unclear why Figure 1 was mentioned in Section 1 but not mentioned again in Section 4).
- I would suggest the authors to be slightly more careful with the wordings. Section 1 in p. 1 mentions "To infer causal structure from data it has been shown that it is necessary to perform interventions", which is not entirely true because, as the authors also noted, there is a large class of causal discovery methods based on purely observational data.
- Similarly, in p. 2, the paper mentions "Almost all aforementioned approaches can not make use of interventions at all or are computationally inefficient". This is arguably not true because there is an entire subfield in causal discovery based on active interventions; see my comments about the literature below.

Comments about literature:
- I was surprised that the entire subfield of causal discovery based on active learning/interventions and experimental design was not discussed, e.g. [3, 4, 5, 6], which is highly relevant to the considered problem setting. The authors should discuss the difference between the existing active learning setting and their proposed meta RL setting.
- Other relevant references for Section 1: [7] for "algorithms that search over the space of permutations"; [8, 9] for "encoder-decoder models"; [10] for "constraint optimization".
- I would also suggest the authors to compare their setting/method to [11].

Comments about experiments:
- Following my comments about the literature, I would suggest the authors to compare their method to the existing active learning/experimental design methods.
- It may be unfair to compare to GES in the experiment as it does not make use of the information that the underlying error variances are identical. The authors should instead compare to methods that handle identical error variances, e.g. [12, 13, 14].

Refs:
1. Structural Intervention Distance (SID) for Evaluating Causal Graphs. Neural Computation, 2013.
2. Multi-domain Causal Structure Learning in Linear Systems. In NeurIPS, 2018.
3. Optimal experimental design via Bayesian optimization: active causal structure learning for Gaussian process networks. arXiv, 2019.
4. Active learning of causal bayes net structure. 2001.
5. Interventions, Where and How? Experimental Design for Causal Models at Scale. arXiv, 2022.
6. Learning Neural Causal Models with Active Interventions. arXiv, 2022.
7. Learning directed acyclic graphs based on sparsest permutations. Stat, 2018.
8. Amortized learning of neural causal representations. arXiv, 2020.
9. A Graph Autoencoder Approach to Causal Structure Learning. arXiv, 2019.
10. DAGs with NO TEARS: Continuous Optimization for Structure Learning. In NeurIPS, 2018.
11. Causal Induction from Visual Observations for Goal Directed Tasks. arXiv, 2019.
12. Identifiability of Gaussian structural equation models with equal error variances. Biometrika, 2014.
13. On the Role of Sparsity and DAG Constraints for Learning Linear DAGs. In NeurIPS, 2020.
14. Optimal estimation of Gaussian DAG models. In AISTATS, 2022.

---

### Official Review · Reviewer_XnTL · 2022-06-29
**Review - lacking rigorous evaluation**

**Rating:** 5
**Confidence:** 3

**Review:**

**Summary**: This paper introduces a reinforcement learning algorithm that learns to perform interventions in order to construct a causal graph. The policy is based on actor critic method with a recurrent LSTM layer, where the reward is a special form of the structural hamming distance (SHD) wrt. to the ground truth causal graph and the actions are either i) intervening on a specific node or ii) changing the current PDAG estimation of the agent.

- Pros:
    - The proposition and the motivation is clear. The authors discuss the goal of efficiently choosing interventions as well as generalising to new environments.
    - The connection between experimental design/intervention targeting and reinforcement learning has been established in previous works. It is a nice idea to use RL with the action space of interventions in contrast to previously used heuristics.
- Cons:
    - While the idea is clearly presented, the empirical evaluation does not seem sufficient to asses the correctness of the work. If I understand correctly, the RL model samples from the underlying SCM in *each* training step. E.g., after 50k training steps, it will thus have seen 50k samples (and possibly as many interventions), whereas the baselines only see 1000 observational samples per test instance. Hence, there is a clear contrast between the compared methods in terms of number of samples as well as learning method (active vs. passive) even if the agent collects data across different SCMs.
    - Moreover, the graphs with d={3,4} are very small instances. There, the SHD largely depends on the sparseness of the graphs (e.g. competitive SHD can be achieved with empty graph predictions), and there is no exact description on what graphs are sampled when creating SCMs (e.g. how many edges)
    - It is also not clear to me how the proposed model is better as the two single baselines with only observational data. Why was there no baseline that can leverage interventional data, e.g. GIES or IGSP?
    - The paper does also not discuss and differentiate from related work [1] that is conceptually very close to the authors’ method.

I therefore suggest that the authors extend and rigorously formulate their experimental evaluation. I believe that the proposed model could then better show its potential, but in the current form it is difficult to advocate for acceptance at this workshop.

[1]: Amir Amirinezhad, Saber Salehkaleybar, Matin Hashemi. Active Learning of Causal Structures with Deep Reinforcement Learning. [https://arxiv.org/pdf/2009.03009.pdf](https://arxiv.org/pdf/2009.03009.pdf)

---

### Meta-Review · Program_Chairs · 2022-07-06

**Recommendation:** Accept (Poster)
**Confidence:** 2

**Metareview:**

The reviewers lamented insufficient experimental evaluation, as well as raising comments on writing quality and certain aspects of the method. Nevertheless, the proposed reinforcement learning algorithm that learns to perform interventions in order to construct a causal graph appears to be of interest for the workshop. The authors are encouraged to add more discussion on related literature as suggested by the reviewers for the camera-ready version, and to strengthen experimental evaluation for future versions of the paper.

---

### Decision · Program_Chairs · 2022-07-06

Accept (Poster)